# Transcriptomic Analysis Insight into the Immune Modulation during the Interaction of *Ophiocordyceps sinensis* and *Hepialus xiaojinensis*

**DOI:** 10.3390/insects13121119

**Published:** 2022-12-05

**Authors:** Xinxin Tong, Ting Peng, Sukun Liu, Daixi Zhang, Jinlin Guo

**Affiliations:** Key Laboratory of Standardization of Chinese Medicine, Ministry of Education, Key Laboratory of Systematic Research, Development and Utilization of Chinese Medicine Resources in Sichuan Province—Key Laboratory Breeding Base of Co-Founded by Sichuan Province, College of Pharmacy, Chengdu University of Traditional Chinese Medicine, Chengdu 610075, China

**Keywords:** *Hepialus xiaojinensis*, *Ophiocordyceps sinensis*, immunological interaction, RNA-seqs

## Abstract

**Simple Summary:**

*Ophiocordyceps sinensis* parasitizes the ghost moth larva (*Thitarodes* spp.) and produces a fruit-body, which is a well-known Traditional Chinese Medicine (TCM). The infection process is time-consuming and inefficient, restricting the large-scale cultivation of this fungus. This study aimed to reveal the biology of immune modulation during the interaction between *O. sinensis* and its host (*Hepialus xiaojinensis)* using high-throughput RNA sequencing. In *H. xiaojinensis,* 345 immune-related gene candidates were identified and were further classified according to the functions of pathogen recognition, signal transduction and immune response. According to the results of the analysis, the putative pathways of immune modulation in *H. xiaojinensis* in response to *O. sinensis* infection were sketched. Meanwhile, gene families probably involved in *O. sinensis* pathogenicity were identified, mainly including serine carboxypeptidase, peroxidase, metalloprotease peptidase, aminopeptidases, cytochrome P450, and oxidoreductase. Our findings provided an insight into the biology of fungi–insect host interaction, paving the way for large-scale artificial cultivation of the valued fungus as well as biological pest control.

**Abstract:**

*Ophiocordyceps sinensis* (Berk.) is an entomopathogenic fungus that can infect the larva of the ghost moth, *Hepialus xiaojinensis,* causing mummification after more than one year. This prolonged infection provides a valuable model for studying the immunological interplay between an insect host and a pathogenic fungus. A comparative transcriptome analysis of pre-infection (L) and one-year post-infection (IL) larvae was performed to investigate the immune response in the host. Here, a total of 59,668 unigenes were obtained using Illumina Sequencing in IL and L. Among the 345 identified immune-related genes, 83 out of 86 immune-related differentially expressed genes (DEGs) had a much higher expression in IL than in L. Furthermore, the immune-related DEGs were classified as pathogen recognition receptors (PRRs), signal modulators or transductors, and immune effector molecules. Serpins and protease inhibitors were found to be upregulated in the late phase of infection, suppressing the host’s immune response. Based on the above analysis, the expression levels of most immune-related genes would return to the baseline with the immune response being repressed in the late phase of infection, leading to the fungal immunological tolerance after prolonged infection. Meanwhile, the transcriptomes of IL and the mummified larva (ML) were compared to explore *O*. *sinensis* invasion. A total of 1408 novel genes were identified, with 162 of them annotated with putative functions. The gene families likely implicated in *O. sinensis* pathogenicity have been identified, primarily including serine carboxypeptidase, peroxidase, metalloprotease peptidase, aminopeptidases, cytochrome P450, and oxidoreductase. Furthermore, quantitative real-time PCR (qPCR) was used to assess the expression levels of some critical genes that were involved in immune response and fungal pathogenicity. The results showed that their expression levels were consistent with the transcriptomes. Taken together, our findings offered a comprehensive and precise transcriptome study to understand the immune defense in *H. xiaojinensis* and *O. sinensis* invasion, which would accelerate the large-scale artificial cultivation of this medicinal fungus.

## 1. Introduction

*Ophiocordyceps sinensis* (Berk.), belonging to the family Ascomycete, is a highly valued traditional Chinese medicine (TCM) fungus that has been used in several Asian countries for over 2000 years [1,2]. This fungus was reported to have over 20 bioactivities including adenosine, cordycepic acid, ergosterol, and polysaccharides, which were thought to be responsible for its numerous pharmacolgical effects, including immunomodulatory, antitumor, and antioxidative activities [2,3]. In recent years, the drastically declining yield of this fungus has been unable to meet the increasing market demand [4,5]. Nevertheless, the large-scale artificial cultivation of *O. sinensis* hasn’t been accomplished due to major obstacles such as low rates of fungal infection, primordium induction, and fruit-body development [5].

*O. sinensis* is an entomopathogenic fungus with a complicated life cycle: it specifically colonizes ghost moth caterpillars (*Thitarodes* spp.), mummifies the larva, and grows a stalked fruiting body from the sclerotium head [1,2]. Unlike most insect pathogens, which grow rapidly in insects and kill them within three to five days of infection, there was a latent period of three to four weeks during which the infected insects continued feeding, and the hemocoel contained few fungal cells. Some studies demonstrated that it can rapidly respond to *O. sinensis* infection at the early phase but the host will tolerate the fungus’s proliferation in the late phase, leading to a chronic infection that can last for more than one year [6]. The prolonged chronic infection was most likely caused by the suppression of both of antimicrobial peptide (AMP) synthesis and essential components of the Toll/immune deficiency (IMD) pathways in the late phase of infection [6]. In terms of fungal pathogenesis, genome analysis revealed that *O. sinensis* has a significant lineage-specific expansion of gene families enriched in fungal pathogenicity, such as peroxidase activity, serine hydrolase, deuterolysin metalloprotease (M35) peptidase, cytochrome P450 (Cyt P450), and glucose-methanol-choline (GMC) oxidoreductase [7,8,9]. The extraordinary expansion and positive selection of numbers of these genes may contribute to the specific host infection [7]. However, the host immunity and *O. sinensis* pathogenesis remain elusive. The long-lasting mutualistic occurrence between this fungus and its host larva provides a model system for scientific research and artificial cultivation. 

In this study, the insect larvae *(Hepialus xiaojinensis*) were successfully reared and infected by *O*. *sinensis* under our laboratory conditions [10]. The RNA-seqs of pre-infected, one-year post-infected, and mummified larvae (L, IL and ML) were comparatively analyzed. Our findings provided a thorough analysis of transcriptome to understand the immunological modulation between the host insect and *O. sinensis* accelerating the large-scale artificial cultivation of this highly valued medicinal fungus, as well as the biological control of insects.

## 2. Materials and Methods

### 2.1. Specimen Collection, RNA Extraction

*H. xiaojinensis* larvae were reared and infected by *O. sinensis* in our lab using the following method. The pupae of the host insects were collected from 3500–4000 m mountains in Xiaojin County, Sichuan province, China, and housed with the roots of Potentilla anserina as food in plastic containers at 9–13 °C and 40–50% relative humidity (RH). Fifty 4–5th instar larvae were collected for *O. sinensis* infection. 

The mature fruit-bodies of *O. sinensis* were harvested in 3000 to 4000 m mountains in Aba Prefecture, Sichuan province, China in May 2020 and stored at 15–18 °C and 75% RH. Aseptic bags were wrapped around the fruiting bodies to collect discharged ascospores. Ascospores were washed from bags with ddH_2_O and diluted to about 1 × 10^6^ spores/μL in potato dextrose agar (PDA) liquid medium (20% potato, 0.3% KH2 PO4, 0.15% MgSO_4_, 2.5% glucose and 0.001% vitamin B) and sprayed uniformly onto the 4–5th instar larvae 2 or 3 times per day, then kept at 13 °C and 40–50% RH. After one year, about 10 μL hemolymph of each infected larva was sampled for the presence of O. sinensis blastospores to confirm that the larva was successfully infected.

Three biological triplicates of fresh IL samples were kept in RNAlater (Ambion, Austin, TX, USA) for RNA extraction prior to use, as shown in Figure 1. Total RNA was extracted from the entire IL by using TRIzol (Invitrogen, Waltham, MA, USA) according to the manufacturer’s instructions. Genomic DNA was digested by DNase I (Fermentas, Waltham, MA, USA). Purified RNA was electrophoresed in a 1% agarose gel, and the purity and the quality of RNA were assessed by OD260 and OD230. After that, at least 20 µg total RNA was then submitted to Biomarker Technology Co., Ltd. (Beijing, China) for quality control using an Agilent Bioanalyzer 2100 (Agilent Technologies, Davis, CA, USA).

The mRNA was enriched from total RNA using poly(T)+ oligo attached magnetic beads, eluted with Tris–HCl buffer and fragmented in fragmentation buffer using an RNA fragmentation kit (Ambion, Austin, TX, USA). First-strand cDNA was synthesized with fragmented mRNA as template and random hexamers as primers, followed by second-strand synthesis with the addition of PCR buffer, dNTPs, RNase H, and DNA polymerase I. Purification of cDNA was processed with AMPure XP beads. Double-strand cDNA was subjected to end repair. Adenosine was added to the end and ligated to adapters. AMPure XP beads were applied here to select fragments ranging in size from 300 to 400 bp. cDNA library was obtained by certain rounds of PCR on the cDNA fragments. 

### 2.2. Sequencing and Assembly of Transcriptome Data 

Sequencing of the cDNA library was performed by Illumina NovaSeq based on Sequencing by Synthesis technology (SBS) at Biomarker Technology Co., Ltd. (Beijing, China). Clean data with high quality was obtained by filtering raw data, removing adapter sequences and reads with low quality (<Q30), with default parameters. In terms of IL comprising both the host and *O. sinensis*, each RNA-seq was assembled separately. For the assembly of RNA-seqs of *O. sinensis*, Hishat2 [11] aligner tool (http://ccb.jhu.edu/software/tophat/index.shtml (accessed on 10 May 2021)) was used to align sequencing reads to the *O. sinensis* reference genome under the accession No. PRJNA608258 in NCBI database. StringTie [12] was applied to assemble the mapped read. For the assembly of RNA-seqs of the host, the clean reads mapped to the *O. sinensis* reference genome were removed and then the left reads were used in the subsequent de novo transcriptome assembly with Trinity (version 2.7.0, Broad Institute, Cambridge, MA, USA) using default settings [13,14]. Finally, De Bruijn [14] was applied to recognize transcripts.

### 2.3. Analysis and Annotation of Differentially Expressed Genes 

Coding sequences were predicted by Transdecoder (version 2.0.1, Commonwealth Scientific and Industrial Research Organization, ACT, Australia) and then were annotated by blasting against Pfam database [15] using HMMER [16]. Functional annotation of assembled sequences was retrieved from the databases of NCBI’s non-redundant database (NR, http://www.ncbi.nih.gov/blast/db/ (accessed on 10 May 2021)), Swiss-Prot (http://www.expasy.org (accessed on 10 May 2021)), Gene Ontology database (GO, http://www.geneontology.org/ (accessed on 10 May 2021)), Cluster of Orthologous Groups (COG, http://www.ncbi.nlm.nih.gov/COG/ (accessed on 10 May 2021)), KOG (http://www.ncbi.nlm.nih.gov/KOG/ (accessed on 10 May 2021)), eggnog (version 4.5, http://eggnogdb.embl.de/ (accessed on 10 May 2021)), Kyoto Encyclopedia of Genes and Genomes (KEGG, http://www.kegg.jp/ (accessed on 10 May 2021)). KEGG Orthology of unigenes was obtained by KOBAS2.0. 

To globally characterize the expression profiles of these RNA-Seq samples, paired-end reads were aligned back to the assembled transcripts using Bowtie 2.0 as the aligner. RSEM, the utility package of the Trinity software, was performed to estimate the abundance of transcripts (http://deweylab.github.io/RSEM/ (accessed on 10 May 2021)) and the fragments per kilobase per million mapped reads (FPKM). Differentially expressed genes (DEGs) were calculated using DEseq2 in R. The expression difference with Fold Change(FC) ≥ 2 and FDR < 0.01 was considered to be significant. 

### 2.4. Identification of Immune-Related Genes from the Comparative Transcriptomes of H. xiaojinensis

The probable immune-related genes in *H. xiaojinensis* were manually selected based on the annotation results. Conserved domain structures were determined using the CD-search tool (http://www.ncbi.nlm.nih.gov/Structure/bwrpsb/bwrpsb.cgi/ (accessed on 15 May 2021)), PROSITE (http://prosite.expasy.org/ (accessed on 15 May 2021)), and SMART (http://smart.embl.de/ (accessed on 15 May 2021)). Signal peptide and transmembrane domains were predicted using the SignalP 4.1 server (http://www.cbs.dtu.dk/services/SignalP/ (accessed on 15 May 2021)) and TMHMM server version 2.0 (http://www.cbs.dtu.dk/services/TMHMM/ (accessed on 15 May 2021)). Protein structure was predicted by AlphaFold2. Multiple sequence alignments were performed using CLUSTALX 2.0 (The Conway Institute of Biomolecular and Biomedical Research, Cambridge, UK). 

### 2.5. Quantitative Real-Time PCR 

Quantitative real-time PCR (qPCR) was performed on a CFX96 Touch qPCR system (Bio-rad, Santa Clara, CA, USA) according to manufacturer’s instructions. Quantitative measurements were carried out in three biological triplicates for each sample. Elongation factor 1 α subunit (gene-G6O67_008216) and 18sRNA were used as an internal standard to normalize the cDNA for the assessment of expression level of transcripts in IL vs. L and IL vs. ML, respectively. All primers used for the selected genes are listed in Appendix A. 2^−∆∆ct^ values were considered to be the relative expression levels or fold change between samples. The results are presented as the mean ± SD. Two-tailed Student’s *t*-tests and graphs were performed using GraphPad Prism software (version 5.0, GraphPad Software) and used for statistical analysis.

### 2.6. Data Availability 

The raw RNA-seqs of L were derived from the NCBI database (https://www.ncbi.nlm.nih.gov/ (accessed on 1 May 2021)) under the accession No. SRR1735514, SRR5282572, SRR5282573. The raw data of ML was derived from the NCBI database under the accessions No. SRX9405963, SRX9405964, SRX9405965. The sequencing reads of IL was submitted to GSA database (https://ngdc.cncb.ac.cn/gsub/ (accessed on 1 July 2022)) under the accession No. subCRA012084 with the BioProject No. PRJCA011311.

## 3. Results and Discussion

### 3.1. RNA Sequencing Analysis

Three biological replicates of IL were submitted for RNA sequencing to reveal the biology of insects’ immune response to *O. sinensis*. The raw data from the sample was processed to compare the expression profiles between L and IL. Following cleaning and quality control, 46.15 Gb clean reads were obtained. The percentage of bases with a quality score of Q30 exceeded 92.04%. For each IL replicate, 76.04–77.99% of reads were successfully mapped (Appendix A). Following sequence assembly, 59,668 unigenes were obtained, with 21,423 of them having lengths longer than 1 Kb. 

The gene expression profiles between IL and ML were compared to reveal the *O. sinensis* pathogenesis. After cleaning and quality control, about 54.19 Gb paired-end, clean reads were obtained with Illumina HiSeq. At least 6.66 Gb of clean data were generated for each replicate, with a minimum of 93.74% of clean data achieving a good quality score of Q30. The clean reads of each sample were mapped to the reference genome using HISAT2 [11] and assembled by StringTie [12]. For each replicate, about 7.78% of the reads were successfully mapped to the *O. sinensis* genome, while about 94% of the reads were mapped to the host’s assembled transcripts, indicating that *O. sinensis* accounted for a minor proportion within IL (Appendix A). Following sequence assembly, 1408 novel genes were identified, with 162 of them annotated with putative function.

### 3.2. Analysis of Gene Differential Expression and Functional Enrichment in the Comparison of IL and L

DEGs were considered statistically significant if the |log2 FC(log2 fold-change)| is more than 2 and FDR is less than 0.001. This threshold resulted in a total of 7194 genes (5905 upregulated, 1244 downregulated) as significant DEGs in IL vs. L (Figure 2, Appendix A), indicating that most genes were upregulated in IL compared to L.

Based on the COG database, 4799 DEGs were enriched in GO categorization, with 4426 upregulated and 373 downregulated in IL compared to L (Appendix A). The DEGs were mainly enriched in the GO categories of ATP synthesis coupled electron transport (GO:0042773), ATP metabolic process (GO:0046034) and immune system development (GO:0002520), etc., in biological process (BP), extracellular region (GO:0005576), membrane part (GO:0044425), integral component of membrane (GO:0016021) and super molecular complex (GO:0099081) in cellular component (CC), catalytic activity (GO:0003824), transporter activity (GO:0015405), signal transduction (GO:0007165), and electron carrier activity and antioxidant activity (GO:0016209) in molecular function (MF). The majority of DEGs enriched in the GO categorization were upregulated in IL compared to L (Figure 3), suggesting that the host’s immunological regulation, the demand of energy, and ROS metabolism would be enhanced in response to *O. sinensis* infection.

Pathway-based analyses were performed to further understand the biological functions of DEGs. 3154 DEGs were enriched in KEGG categories. The KEGG pathway enrichment analysis revealed that the upregualted DEGs in IL versus L were highly associated with several pathways, such as carbon metabolism (ko01200), fatty acid metabolism (ko01212), biosynthesis of acid amino (ko01230), ABC transporters, and peroxisome (ko04146). The results showed that extra energy, compounds, and unique proteins might be produced as a defense against *O. sinensis* infection (Figure 4, Appendix A). Fatty acids are significant energy sources, cell membrane structure components, signaling molecules, and precursors for the synthesis of eicosanoids and related mediators [17]. Fatty acids can modulate the behavior of a variety of proteins involved in immune cell activation, including those associated with T cell responses, antigen presentation, and fatty acid-derived mediators [18]. In this study, some critical genes involved in fatty acid metabolism were found to be highly upregulated in IL compared to L, such as polyketide synthase dehydratase (BMK_Unigene_041672), β-ketoacyl synthase (BMK_Unigene_035362), thiolase (BMK_Unigene_022220), and ubiquitin carboxyl-terminal hydrolase (BMK_Unigene_052423). 

Furthermore, 77 DEGs annotated were found to be enriched in the KEGG pathway of peroxisome (ko04146); 84.2% of these DEGs were upregulated in IL compared to L, including Pex2_Pex12 domain-containing protein genes (BMK_Unigene_014313), peroxisomal acyl-coenzyme A oxidase (BMK_Unigene_054313), and cytochrome b2 (BMK_Unigene_044383) (Appendix A). Peroxisomes are thought to be a novel branch of immune metabolism since they have been recognized as crucial regulators of immunological activities and inflammation during infection [19]. Peroxins are proteins that are encoded by the PEX genes and are necessary for peroxisome biogenesis [20]. Peroxisomes are necessary for the resolution of microbial infections via canonical innate immune pathways [21]. In this study, PEX 1/2/5/12/13/14/16 genes were found to have higher expression levels in IL compared to L, suggesting that peroxisome metabolism would play an important role in the host’s immune response to *O. sinensis* infection. A previous study found that the treatment of macrophages with peroxisome-derived lipid mediators enhanced their capacity to engulf bacteria [22]. Here, a total of 30 DEGs implicated in peroxisome-derived lipid transport and metabolism were upregulated in IL compared to L, such as fatty acyl-CoA reductase, AB hydrolase-1 domain-containing protein (BMK_Unigene_043616), and thiolase (BMK_Unigene_059853). Furthermore, the innate immune response is coupled with autonomous or non-autonomous cellular signaling via the production of ROS and cytokine [21]. Here, some genes involved in ROS metabolism, such as superoxide dismutase [Cu-Zn](BMK_Unigene_041997, superoxide dismutase [Mn] (BMK_Unigene_041997) and catalase genes (BMK_Unigene_045733, BMK_Unigene_051539), were found to be upregulated in IL compared to L. The analysis showed that peroxisome might play an important role in the host immune response by regulating the lipid metabolism and ROS levels.

### 3.3. Immune-Related DEGs Analyses 

A total of 345 immune-related genes were annotated, with 86 of them screened for DEGs with log2 FC > 2, FDR < 0.01. The heatmap of immune-related DEGs was shown in Figure 5, indicating that most immune-related genes have no differential expression and immune tolerance would develop in the late phase of infection. Furthermore, 83 of 86 DEGs were found to be upregulated in IL compared to L. These genes were grouped according to their functions in terms of pathogen recognition, signal modulation, and transduction. Pathogen-associated molecular patterns (PAMP) bind to PRRs on the surface of microbes and induce phagocytosis as well as downstream immune responses [23]. Upon PAMP recognition, PRRs rapidly trigger an array of anti-microbial immune responses through the activation of various inflammatory cytokines, chemokines, and type I interferons [23]. Here, 29 PRRs were identified, including nine peptidoglycan recognition proteins (PGRP), 11 β-1,3-glucan recognition protein (βGRP), four C-type lectins (CTL), and five scavenger receptors (SRs) (Appendix A). 

A PGRP was initially isolated from the hemolymph of the silkworm, *Bombyx mori* [24]. When PGRP binds to peptidoglycan (PGN), it activates the prophenoloxidase cascade and signal transduction pathways, which lead to the production of immunological effectors [24]. Drosophila, mosquitoes and mammals have families of thirteen, seven, and four PGRP genes, respectively, and some of these genes are alternatively spliced in *Drosophila* [25]. In this study, nine genes were annotated to PGRPs. The PGRP family has a PGRP domain in C-terminal region that is homologous to the T4 bacteriophage lysozyme and is the key element that can recognize bacterial PGN and is conserved from insects to mammals [26]. In this study, the analysis of the PGRP protein structure showed that there are two types of PCRPs: long-type PGRPs (PGRPs-L), which include PGRPs -LC and -LB > 90aa, 1–2 transmembrane regions, and a PGRP domain, and short-type PGRPs (PGRPs-S), which include PGRPs-SC, PGRPs-SA, a signal peptide region, a PGRP domain, and no transmembrane regions (Figure 6). It is indicated that PGRPs-S might be secreted to the extracellular space and PGRPs-L may serve as a receptor on the surface of cells and/or as an intercellular receptor. The key residues of PGRPs-LB (Trp394, Asp395, His365, Glu480, Ser477, Ala478, and Thr479) were found to be required for the capacity to bind to *meso*-2,6-diaminopimelic acid (DAP)-type PGNs, enabling them to defend against some gram-negative and -positive bacteria [26]. PGRP-LE lacks a signal peptide and locates in the hemolymph and cytoplasm of immune cells [27]. It is a multifunctional pattern recognition molecule that activates both of IMD pathways and proPO cascade [27]. Our data showed that most PGRPs exhibited no differential expression, and two PGRP-LB genes were downregulated in IL compared to L, indicating that PGRPs might function in immunological recognition in the early phase of infection. Furthermore, the amino acid sequences of βGRPs revealed that they had a glucan-binding domain for β-1, 3-glucan on the fungal surface (about 100 aa) and a β-glucanase domain (about 290 aa), as shown in Figure 6. The conserved catalytic residues (Glu, Asp, and Glu) involved in maintenance of the glucanase activity were also found in βGRP 3 of H. xiaojinensis. in response to fungal infections, βGRP 3 is found to be required for the activation of the Toll pathway [28]. Here, βGRP 1-3 were identified in H. xiaojinensis and contained a potential signal peptide, suggesting that they might be secreted into the hemolymph. A total of 11 βGRPs transcripts were identified, with three βGRPs 1 being highly upregulated in IL compared to L. A previous study showed that βGRP 3 exhibited a quick response within 12 h after infection but the other βGRPs showed strong induction at 72 h after infection [29]. Hence, it was suggested that βGRPs 1 might have a role in the host’s immune response in the late phase of infection. 

C-type lectin (CTL) serves as a key PRR in innate immunity by recognizing a wide spectrum of pathogens [30]. In this study, four C-type lectin genes were identified in *H. xiaojinensis*, including CTL 5, CTL 6, and immulectin-21 and -22, which bind specifically to certain sugars and cause agglutination of particular cell types. In *B. mori*, CTL 5 might be an important PRR for activating the JAK/STAT signaling pathway and mediating the nodule melanization for fungal infection [30]. In *B. mori*, CTL-S6 may recognize foreign pathogens, activate the prophenoloxidase pathway and innate immune response [30]. In this study, two CTLs were found to be upregulated in IL compared to L, suggesting that they might function as PRRs to trigger immune response in *H. xiaojinensis*. SRs are also an important subclass of PRRs and have a ligand-binding specificity [31]. In this study, five SRs were identified in *H. xiaojinensis*. A higher expression level of the two transcripts in IL compared to L showed that SRs might also participate in activating immune response in the late phase of infection.

Clip domain serine proteases (CSPs) in insect hemolymph are rapidly activated by specific proteolysis after pathogen recognition and participate in extracellular signal amplification and countervailing serine proteases [32]. Here, 35 CSPs transcripts were identified in *H. xiaojinensis* and even more in other Lepidopteran insects, indicating that CSPs would play a key role for the host immunity [29]. Microbial infection can stimulate the activities of CSPs and prophenoloxidase [32,33]. CSPs activate the Toll signal pathway and then produce the downstream immune effectors [33]. The analysis of CSPs protein sequences showed that they comprised of a signal peptide (SP), 1–2 clip domains, and one serine protease domain (Figure 7A,B). In the clip domain, four conserved Cys residues were arranged in two disulfide-bridge structures, which are responsible for signal amplification by cleaving and activating the downstream signals (Figure 7A,C). The serine protease domain with about 37–55 aa at the N-terminus, containing six conserved Cys residues that were arranged in three disulfide-bridge structures, had the activity of serine protease, as shown in Figure 7A,C. The clip domain was interlinked by three strictly conserved disulfide bonds and connected to the SP domain through a linker of about 23–92 aa (Figure 7B,C). In this study, 26 CSPs transcripts were identified in *H*. *xiaojinensis*, including prophenoloxidase-activating enzyme 3, CSPs 2, and melanization protease 1. ModSP, a modular serine protease, plays an essential part in the activation of the Toll pathway by gram-positive bacteria and fungi [34]. In this study, four ModSP were identified and were found to be upregulated in IL compared to L. Furthermore, the Spz-processing enzyme (SPE)-mediated process is modulated by serine protease family in *H*. *xiaojinensi* [29]. On the contrary, serpins regulate the innate immune responses of insects by inhibiting their cognate SPs [35]. They act as bait for their target proteases via an exposed C-termination [36]. A previous study found that serpins were found to be upregulated in the late phase of infection in *H*. *xiaojinensis* [29]. In this study, 43 serpin transcripts were identified; two of them were highly upregulated in IL compared to L, while the others had no differential expression, similar to the previous study [29]. Based on the above analysis, the immunity response would be suppressed in the late phase of infection in *H*. *xiaojinensis*.

Mitogen-activated protein kinases (MAPKs) also regulate immune response by producing immunomodulatory cytokines [37]. Following pathogenic infection, the activation of PRRs on the surface and in the cytoplasm of immune cells activates members of MAPK subfamilies—the extracellular signal-regulated kinase (ERK), p38, and Jun N-terminal kinase (JNK) subfamilies [38]. Toll-like receptor TLR3 induces signaling pathways involving TRIF, p38 MAPK, and MK2 to enhance the IFN-β levels during pathogen infections [39]. In this study, four DEGs involved in Toll/IMD signaling pathways (ko04624) and MAPK- (ko04013) were found to be upregulated in IL compared to L, including mitogen-activated protein kinase kinase kinase A (MAPKKKA), MAPK and RanBP2-type domain-containing protein, suggesting that the Toll/IMD coupling with MAPK pathway would regulate the immune innate in *H. xiaojinensis*. Besides, lysosome-autophagy is the first-line defense against pathogens. Autophagy defends against pathogenic infection as a part of the cellular defense system by producing autophagosomes that transport these cargoes to lysosomes for degradation [40]. In this study, three transcripts implicated in the KEGG pathway of lysosome, including MD-2-related lipid-recognition protein and serine carboxypeptidase, were shown to be highly upregulated in IL compared to L. Based on above analysis, the potential pathways of the immune defense in *H*. *xiaojinensis* were sketched (Figure 8).

### 3.4. Differential Gene Expression and Functional Enrichment Analysis in the Comparison of IL and ML

DEGs were considered statistically significant if the |log2 FC (log2 fold-change)| is more than 2 and the FDR is less than 0.001. This threshold resulted in a total of 3243 genes (1507 up-regulated, 1736 down-regulated) as significant DEGs in ILvs. L (Figure 9A, Appendix A). Based on the COG database, 3243 DEGs were enriched in GO classification, with 1507 DEGs upregulated and 1736 DEGs downregulated in IL compared to ML (Figure 10). The DEGs were mainly enriched in the GO categories of the metabolic process, signaling, reproductive process and developmental process, response to stimulus, etc., in BP; membrane part, organelle part, macromolecular complex, and extracellular region, etc., in CC; and catalytic activity, antioxidant activity transporter activity, and signal transduction, etc., in MF (Figure 10, Appendix A). Oxidoreductase genes enriched in the GO category of antioxidant activity were mostly highly upregulated in IL compared to ML. For example, Pks11, a highly reducing polyketide synthase, is responsible for the synthesis of an array of products, such as antibiotics and mycotoxins [41]. It plays vital roles in the asexual development, cell wall integrity, and fungal responses to oxidation and UV irradiation of *B. bassiana* [41]. FAD-dependent monooxygenase is a gene cluster that mediates the synthesis of the bibenzoquinone oosporein, a metabolite necessary for fungal pathogenicity that serves to increase fungal proliferation by evading host immunity [42]. Furthermore, the DEGs enriched in GO categories of reproductive process, developmental process, and growth were upregulated in ML compared to IL, indicating that *O. sinensis* would rapidly grow after successfully colonizing the host.

To further understand biological functions of the DEGs, pathway-based analyses were performed. A total of 1498 DEGs were found to be enriched in KEGG categories. Among them, the fatty acid biosynthesis (ko00061), pentose and glucuronate interconversion (ko00040, ko00260), glycosphingolipid biosynthesis (ko00603, ko00604), glycine, serine and threonine metabolism (ko00260), and taurine and hypotaurine metabolism (ko00430), etc., were enriched and upregulated in IL compared to ML, indicating that these pathways might play crucial roles in *O. sinensis* pathogenicity (Figure 11, Appendix A). For example, a variety of glycosphingolipid molecules were expressed on pathogenic fungi [43]. Glycosphingolipids (GSLs) lipid rafts on cellular membranes play an important role in innate and adaptive immunity [44]. Mycriocin, which *Isaria sinclairii* produces in the first step of sphingosine biosynthesis, acts as an antibiotic and immune suppressant, allowing for effective colonization [45]. So, lipid metabolism and its metabolites would play crucial roles in *O. sinensis* pathogenicity. Furthermore, hypotaurine 2, an organic osmolyte or a cytoprotective agent, acts as an antioxidant to scavenge reactive hydroxyl radicals [46]. After infection by pathogens, the host would produce a large amount of ROS; meanwhile, antioxidants would be produced to maintain the balance of ROS [7]. Here, hypotaurine metabolism was upregulated in IL compared to ML, suggesting that ROS antioxidant defense system might be induced during the infection process. Furthermore, genetic studies showed that some serine/threonine kinases and proteases play a role in the infection process [47]. Here, glycine, serine, and threonine metabolism were also upregulated in IL compared to ML, suggesting that it would probably play a role in the pathogenicity of *O. sinensis*.

### 3.5. Analysis of the Gene Families Involved in Fungal Pathogenicity

The *O*. *sinensis* genome showed a considerable expansion of gene families that are mainly involved in fungal pathogenicity, including peroxidase activity, serine hydrolase, deuterolysin metalloprotease (M35) peptidase, Cyt P450, and glucose-methanol-choline (GMC) oxidoreductase involved in the ecdysteroid metabolism of molting in insects [7,48]. These gene families were thought to be related to fungal pathogenicity (listed in Table 1 and Appendix A), and the heatmap of these DEGs in IL vs. ML was shown in Figure 12.

In this study, 37 DEGs were enriched in the Pfam of Cyt P450. Of them, 13 genes were upregulated in IL compared with ML, including Cyt P450 monooxygenase sdnE, Cyt P450/NADPH reductase, and Cyt P450 monooxygenase ABA1. Cyt P450, membrane-bound hemoproteins, play pivotal roles in the detoxification of xenobiotics, cellular metabolism, and homeostasis [49]. CytP450/NADPH reductase is required for electron transfer from NADP to Cyt P450 in microsomes [50]. Furthermore, Cyt P450 enzymes are a major oxidizing agent, causing the constant production of ROS [49]. So, Cyt P450 might be involved in *O. sinensis* pathogenicity via ROS pathway. 

Classically, protease is assumed to be a destructor of host tissues while simultaneously providing nutrients for pathogen replication [51]. Aspartyl protease, a specific extracellular proteolytic system, enables fungi to survive and penetrate tissues [51]. In this study, six DEGs were annotated in the Pfam category of aspartyl protease and mostly downregulated in IL compared to ML, suggesting that the fungal pathogenicity might be suppressed by the host immunity in the late phase of infection. Aminopeptidases involved in the degradation of insect neuropeptides have also been examined in some respects [52]. Among the first proteases found in tissues, metallo-aminopeptidases (MAPs) were exploited in the development of antibacterial, antifungal, and optimal structures that should be evaluated as potential leads in the drug discovery against endogenous and infectious diseases [53,54]. A previous study showed that M35 family was expanded in *O. sinensis*, compared to that in other entomopathogenic fungi [7]. Fungalysins, a family of metalloprotease M36, appears to mediate some aspects of the host–fungus interaction in *A. fumigatus* and some basidiomycetes [55]. In this study, two DEGs (gene-G6O67_000430, gene-G6O67_002713) in the fungalysins family were found to be upregulated in IL compared to ML. Furthermore, the serine carboxypeptidase-like protein (SCPL) family plays a vital role in stress response, growth, development, and pathogen defense [56]. Serine carboxypeptidase is lethal to a subpopulation of flies and suppresses the upregulation of antimicrobial peptides and phagocytosis [57]. In this study, one gene (gene-G6O67_005877) involved in the Pfam category of the SCPL family was also highly expressed in IL compared to ST. Thus, these enzymes might be related to *O. sinensis* pathogenicity.

### 3.6. The Results of qPCR Analysis

To confirm gene expression levels in RNA-seq, some critical genes were selected for qRT-PCR validation. The gene-specific primers used in qRT-PCR are listed in Appendix A. According to the results of qPCR, in IL vs. L, except for one gene (BMK_Unigene_052152,Cyt b5-related protein), the expression levels of the other genes detected by qPCR were consistent with that of the RNA-seqs (Figure 13A,C), including Cyt c (BMK_Unigene_036928), serine protease easter (BMK_Unigene_047872), and Peroxin-14 (BMK_Unigene_047499). These genes related to immunity in *H. xiaojinensis* were detected to be upregulated in IL compared to L. In IL vs. ML, all the qPCR results of all the seven genes were consistent with that of RNA-seq (Figure 13B), including extracellular metalloproteinase 5 (gene-G6O67_002713), serine aminopeptidase (gene-G6O67_007546), and Chitinase 1 (gene-G6O67_003586). These genes most likely related to fungal pathogenicity were detected to be upregulated in IL compared to ML, indicating that these genes would be involved in *O. sinensis* pathogenicity.

## 4. Conclusions

In this study, the immune response of *H*. *xiaojinensis* and the pathogenicity of *O. sinensis* were studied during the infection process by using high-throughput RNA-seq. A total of 345 immune-related genes were identified in *H. xiaojinensis.* Most of these genes had no differential expression. However, serpins and protease inhibitors were found to be upregulated in the late phase of infection. It was indicated that the fungal immune tolerance would develop in the late phase of infection in the host. Furthermore, gene families probably implicated in *O. sinensis* pathogenicity were also identified, primarily including peroxidase, serine hydrolase, serine carboxypeptidase, an aspartyl protease, M35 peptidase, and Cyt P450. Our findings revealed the biology of the immune modulation in the long-term chronic infection process between *H*. *xiaojinensis* and *O. sinensis* using a comprehensive and precise transcriptome analysis.

## Figures and Tables

**Figure 1 insects-13-01119-f001:**
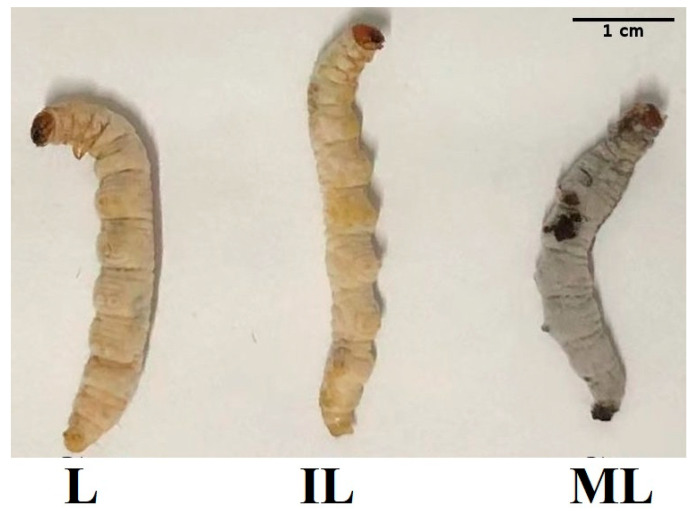
*O. sinensis* samples collection for transcriptome sequencing. IL represents one-year post-infected larva of *H*. *xiaojinensis* by *O. sinensis*. L represents the pre-infected larva of *H*. *xiaojinensis*. ML represents the mummified larva of H. xiaojinensis. Scale = 1 cm.

**Figure 2 insects-13-01119-f002:**
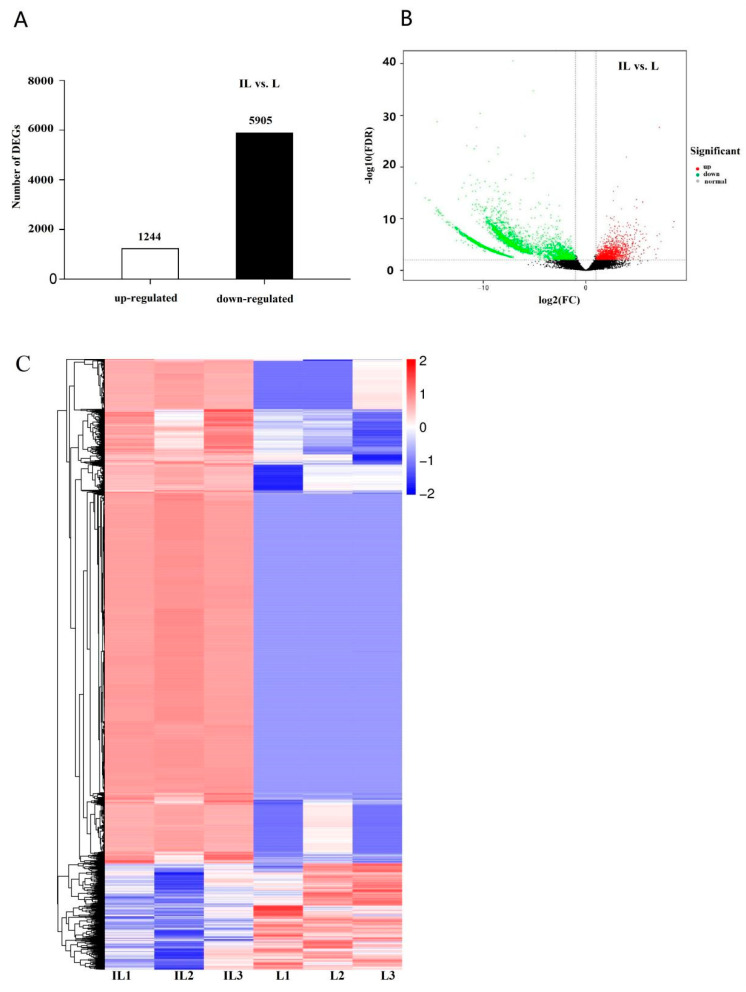
The analysis of differentially expressed genes (DEGs) in IL and L. The histograms of DEGs in groups. (**A**). The number of DEGs is shown on the top of histograms. (**B**). The volcano map of the DEGs in IL vs. L, with each point indicating the differentially expressed genes and the abscissa representing the logarithm of the expression multiple of a gene in the two groups. Green represents downregulation of gene expression in L compared to IL, red represents upregulation of gene expression in L compared to IL, and black dots represent genes with no significant expression differences. (**C**). Hierarchical clustering of differentially expressed genes. Each column represents one sample. Rows represent different genes. The expression level of genes (FPKM) was normalized by log2, and blue color represents the downregulated gene, while red color represents the upregulated genes. L represents the pre-infected larva of *H*. *xiaojinensis*. IL represents the one year post-infected larva of *H*. *xiaojinensis* by *O. sinensis*.

**Figure 3 insects-13-01119-f003:**
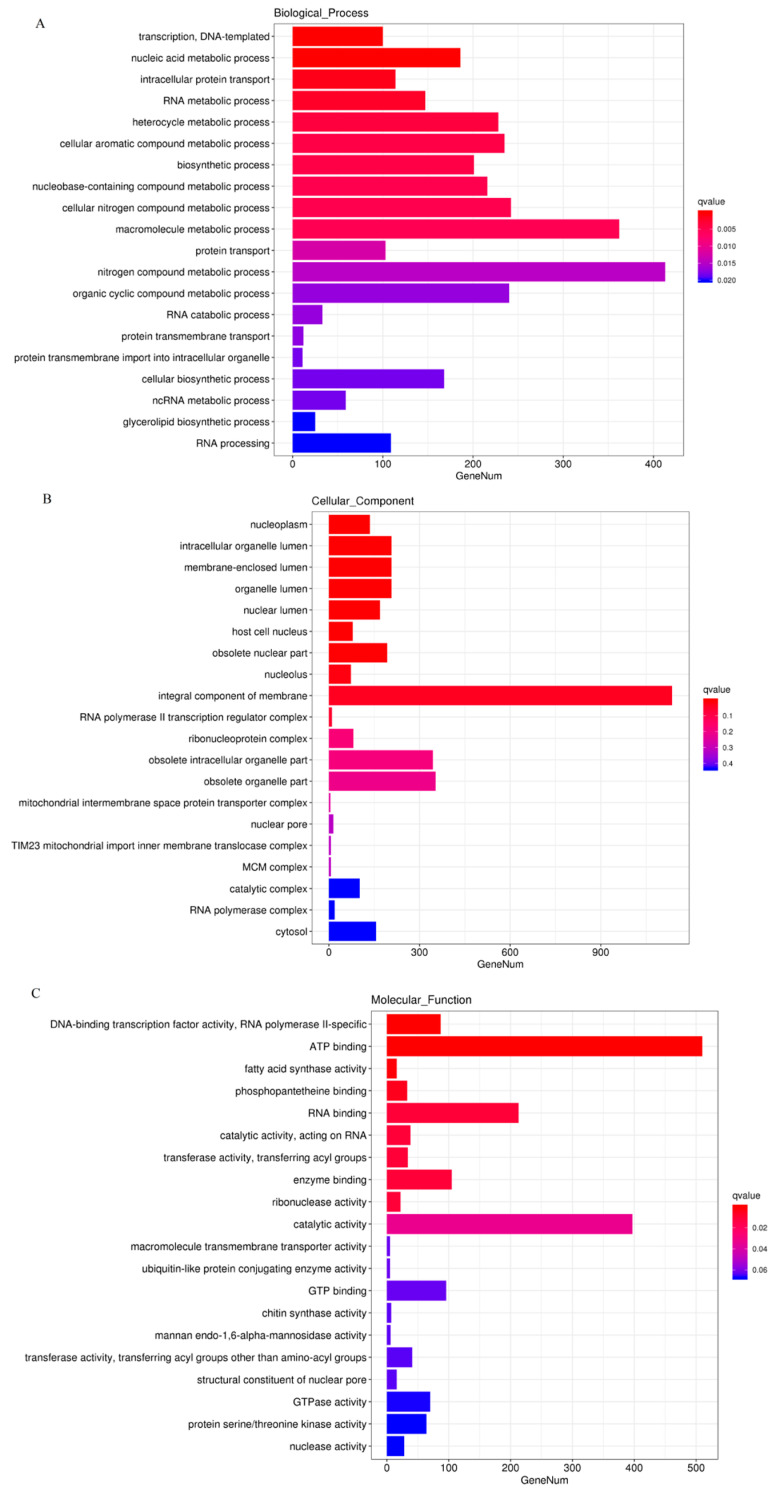
The most enriched GO functional classification of DEGs in IL vs. L. The most enriched GO terms in Biological_Process (**A**), Cellular_Component (**B**) and Molecular_Function (**C**) were presented. *X*-axis represents the gene number of top GO terms enriched among DEGs. L represents the pre-infected larva of *H*. *xiaojinensis*. IL represents the one-year post-infection larva of *H*. *xiaojinensis* by *O. sinensis*.

**Figure 4 insects-13-01119-f004:**
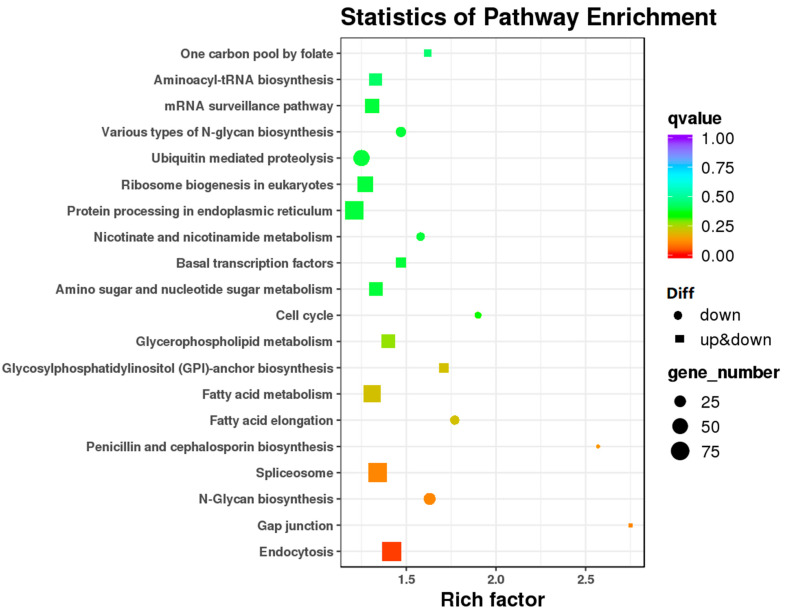
The scatter diagram of the enriched KEGG pathway of DEGs in IL vs. L. *Y*-axis: pathway; *X*-axis: Enrichment factor. The color of the dots stands for q-value. ‘Diff’’ represents differential expression. Circle represents the downregulated DEGs in this pathway, and rectangle represents the upregulated and downregulated DEGs in this pathway. The size of the dots represents the number of DEGs enriched in this pathway. IL represents the one-year post-infected larva of *H*. *xiaojinensis* by *O. sinensis*. L represents the pre-infected larva of *H*. *xiaojinensis*.

**Figure 5 insects-13-01119-f005:**
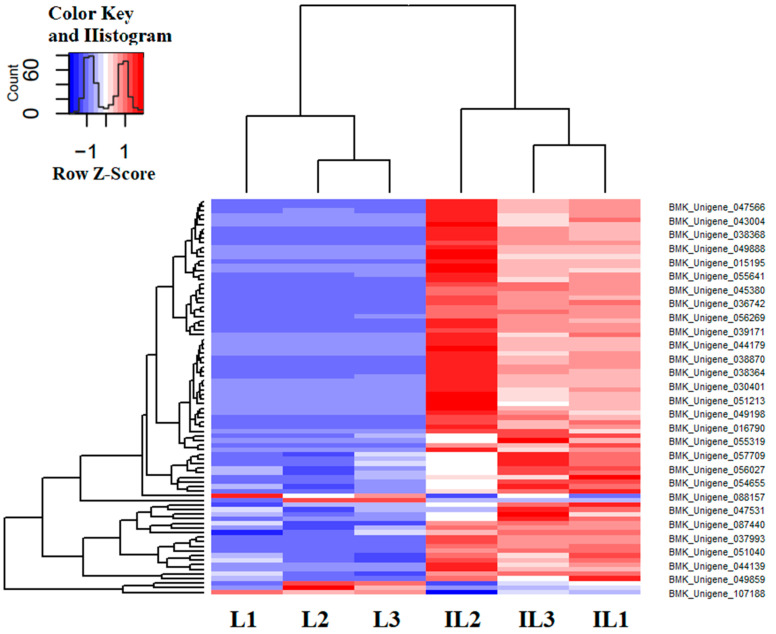
The heatmap of the cluster diagram of differentially expressed immune-related gene expression patterns in IL vs. L. Each column represents one sample. Rows represent different genes. The expression level of genes (FPKM) was normalized by log2 and presented as different colors based on scale bar. L represents the pre-infected larva of *H*. *xiaojinensis*. IL represents the one-year post-infected larva of *H*. *xiaojinensis* by *O. sinensis*.

**Figure 6 insects-13-01119-f006:**
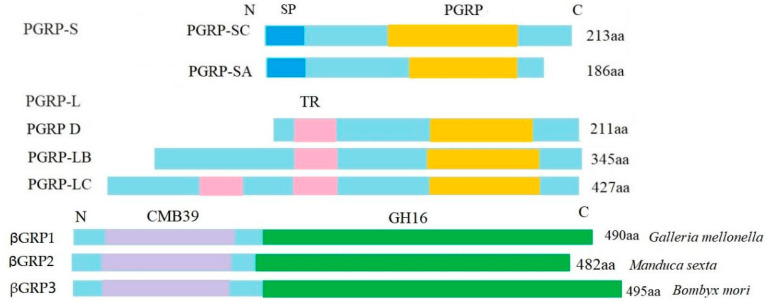
Schematic structure of *H. xiaojinensis* PGPRs and βGRPs identified in the IL and L. The sequence of βGRP 1-3 in *H*. *xiaojinensis* is truncated at the N-terminus. The putative signal peptide (SP), otransmembrane domain (TR), PGRP homologous domain (PGRP), β-1,3-glucan binding domain (CM39), and β-glucanase-like domain (CH16) are indicated in different colored boxes. Lengths of the amino acid sequences are indicated. PGRP-L represents long-type PGRP. PGRP-S represents short-type PGRP. L represents the pre-infected larva of *H*. *xiaojinensis*. IL represents the one-year post-infected larva of *H*. *xiaojinensis* by *O. sinensis*.

**Figure 7 insects-13-01119-f007:**
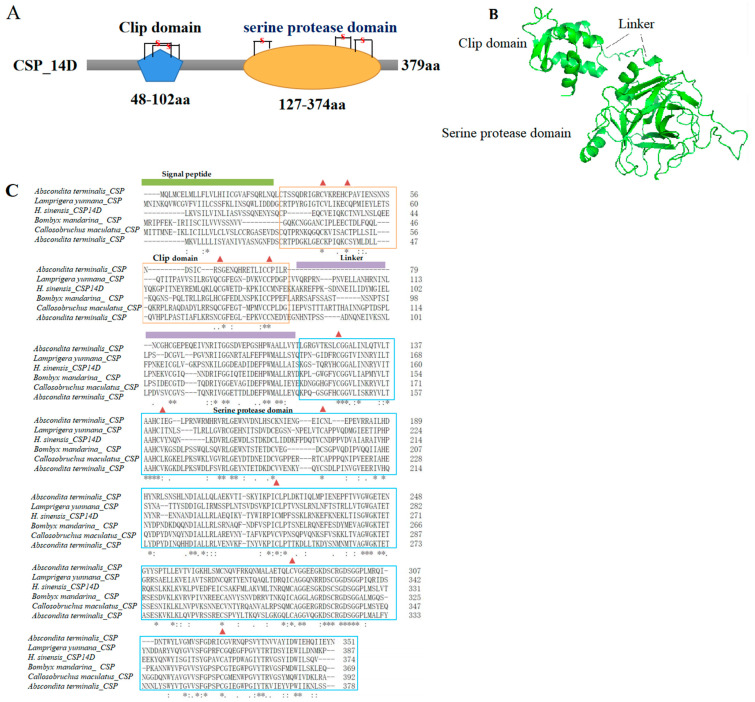
(**A**) Schematic structure of *H. xiaojinensis* CSP_14D. (**B**) Crystal structure of *H. xiaojinensis* CSP (BMK_Unigene_108359) predicted by AlphaFold 2. (**C**) Multiple alignments of the amino acid sequence of CSP with homologs from other insect species. Grey ‘*’ represents 100% identity, grey ‘:’ represents ≥ 70% identity, grey ‘.’ represents 50% identity. Clip domains and serine protease domains are in orange and blue frames, respectively. Green strip represents signal peptide region. Purple strip represents linker region. The conserved cysteine residues responsible for disulfide bonds are labeled with red triangles. CSP: Clip domain containing serine protease.

**Figure 8 insects-13-01119-f008:**
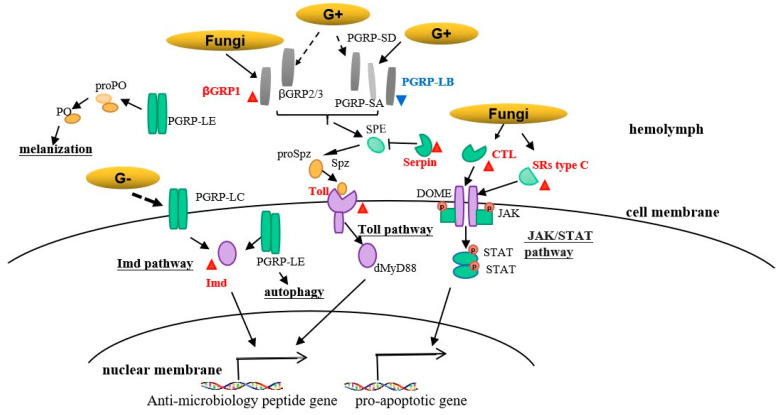
The putative pathways, on the basis of Drosophila research, mediate the response to pathogen infection. The component of the fungal cell wall is sensed by circulating βGRP, peptidoglycan recognition proteins (PGRPs, e.g., PGRP-SA, PGRP-SD and PGRP-LB), a process that activates a proteinase-signaling cascade followed by cleavage of the proSpz (ProSpaetzle) to mature Spz (Spaetzle), binds with Toll, which causes aggregation of intracellular proteins, e.g., MyD88, leading to expression of anti-microbiology peptide (AMP) genes; with the response to the pathogen components, C type lectin (CLT) and/or scavenger receptors (SRs type C) activates JAK/STAT signaling pathway, leading to the expression of pro-apoptotic genes; PGRP-LE in hemolymph activates the prophenoloxidase (proPO) cascade followed by cleavage of the proPO to mature phenoloxidase PO, leading to the melanization for fungi infection; PGRP-LE in cytoplasm activates autophagy-mediated immunity defense; with the responses to fungal infection, PGRP-LC activates Imd pathway, leading to the expression of AMP genes. ‘-’ indicates the name of pathway, the red triangles and letters represent ‘upregulated’ proteins in IL compared to L; the blue triangle and letters represent the ‘downregulated’ protein in IL compared to L.

**Figure 9 insects-13-01119-f009:**
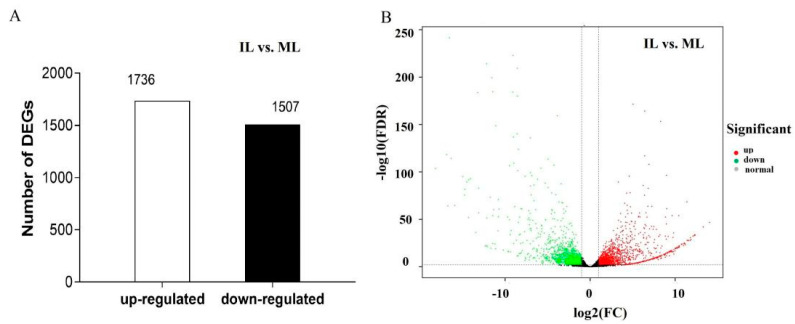
(**A**). The histograms of DEGs in groups; the number of DEGs is shown on the top of histograms. (**B**). The volcano map of DEGs in IL vs. ML; each point represents the differentially expressed gene and the abscissa represents the logarithm of the expression multiple of a gene in the two groups. Green and red dots represent genes with significant expression differences, green represents downregulation of gene expression in ML compared to IL, red represents upregulation of gene expression in ML compared to IL, and black dots represent genes with no significant expression differences. IL represents the one − year post − infection larva of *H*. *xiaojinensis* by *O. sinensis*. ML represents the mummified larva of *H*. *xiaojinensis*.

**Figure 10 insects-13-01119-f010:**
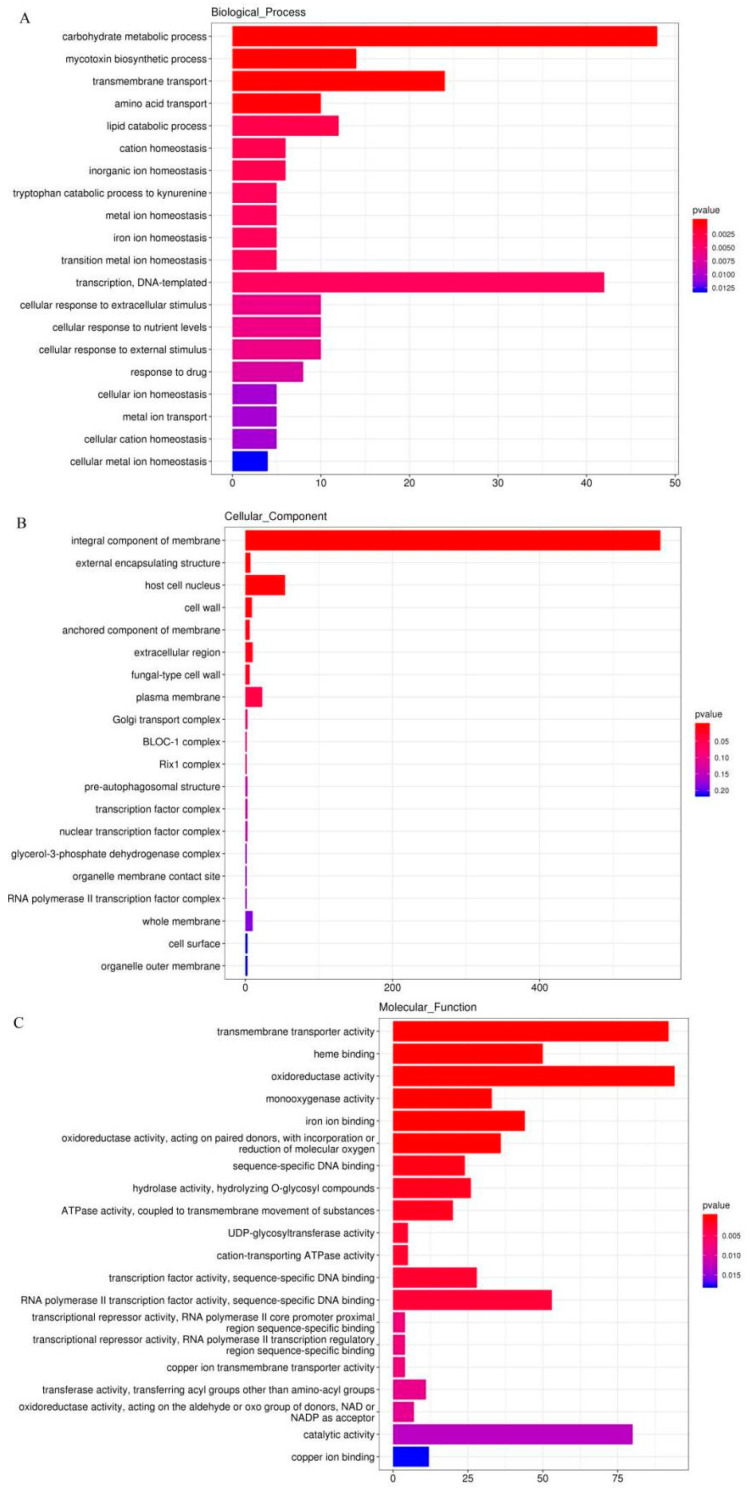
(**A**–**C**) The map of the GO classification of DEGs in IL vs. ML. *X*-axis is the percentage of the genes annotated to this term in total number of annotated genes. The *Y*-axis stands for GO term. IL represents the one-year post-infected larva of *H*. *xiaojinensis* by *O. sinensis*. ML represents the mummified larva of *H*. *xiaojinensis*.

**Figure 11 insects-13-01119-f011:**
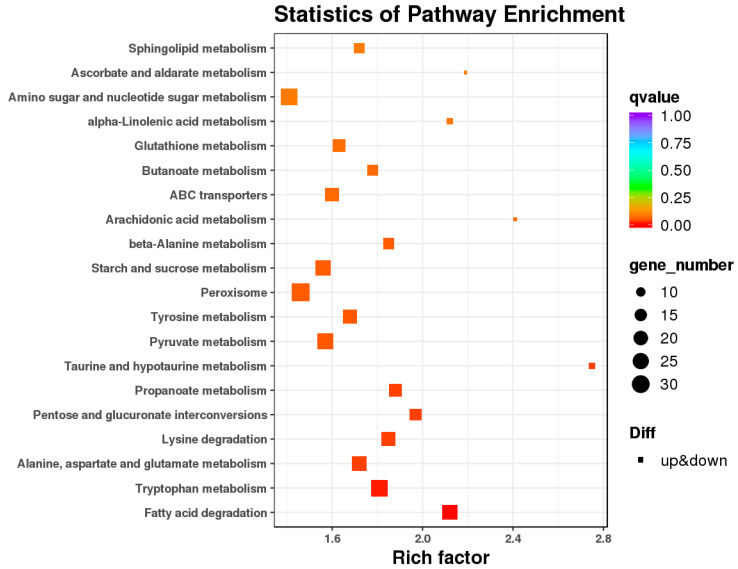
The Scatter diagram of the enriched KEGG pathway of DEGs in IL vs. ML. axis. Pathway; *X*-axis: Enrichment factor. Enrichment factor is calculated as ratio of DEGs annotated to the term over all DEGs. The color of the dots stands for q-value. ‘Diff’ represents differential expression. Rectangle represents the upregulated and downregulated DEGs in this pathway. IL represents the one-year post-infected larva of *H*. *xiaojinensis* by *O. sinensis*. ML represents the mummified larva of *H*. *xiaojinensis*.

**Figure 12 insects-13-01119-f012:**
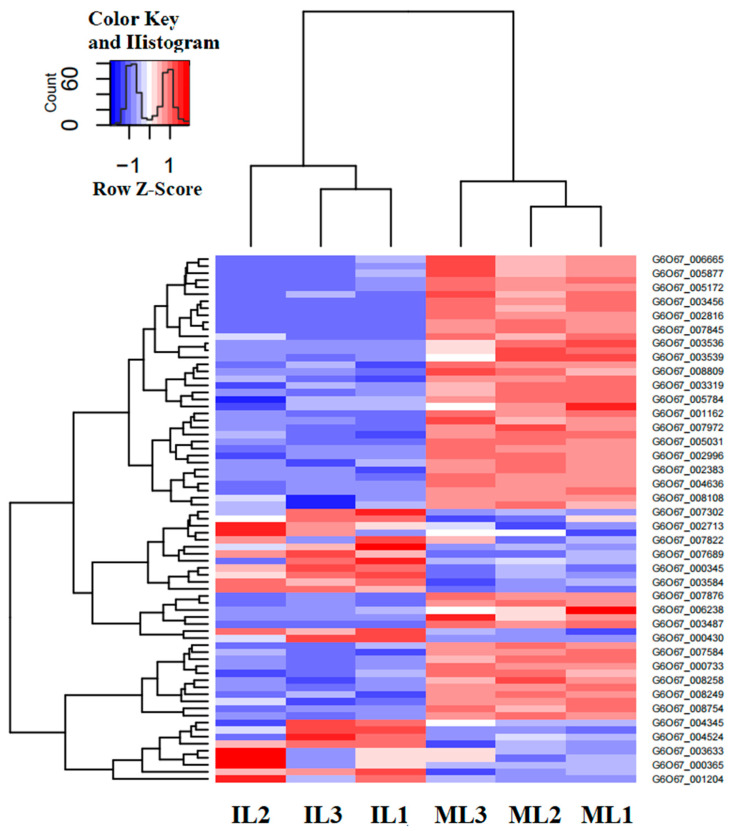
The heatmap of the Cluster diagram of DGEs probably involved in fungal pathogenicity in IL vs. ML. Each column represents one sample. Rows represent different genes. The expression level of genes (FPKM) was normalized by log2 and presents as different colors based on scale bar. IL represents the one-year post-infected larva of *H*. *xiaojinensis* by *O. sinensis*. ML represents the mummified larva of *H. xiaojinensis*.

**Figure 13 insects-13-01119-f013:**
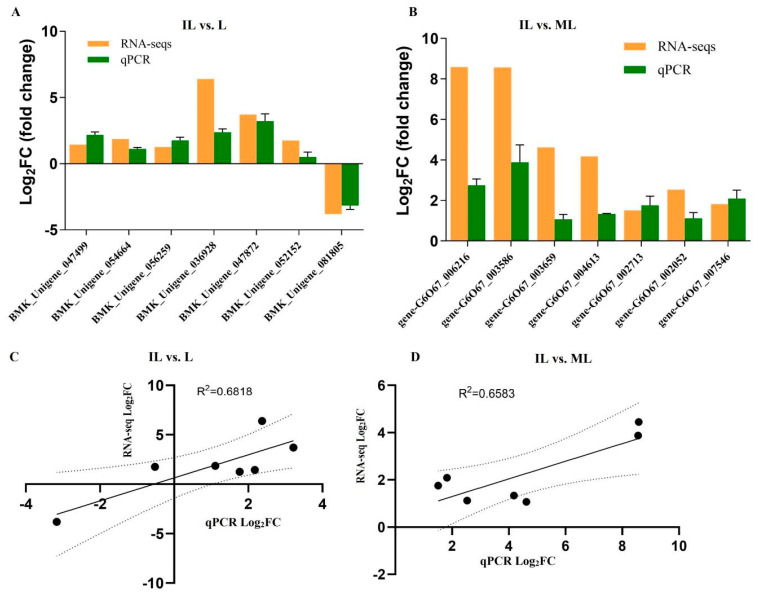
qPCR analysis using gene − specific primers was performed with cDNA samples of L, IL and ML. *H*. *xiaojinensis* larvae were *O. sinensis*-challenged and collected in one year after the infection, as indicated. (**A**,**B**): Log2 fold change comparison with standard deviation, Elongation factor protein gene was used as an internal control to normalize the amount of template in IL vs. L in A. 18s rRNA was used as an internal control to normalize the amount of template in IL vs. ML in (**C**,**D**): correlation analysis between RNA-seq and qPCR log 2 (fold-change) results from the same RNA samples (*p* value = 0.022 and 0.0266 in IL vs. L and IL vs. MF, respectively). Dots represent relative expression of genes, lines represent the regression lines, dash lines represent 95% confidence interval. These experiments were implemented at least in triplicate. L represents the pre-infected larva of *H*. *xiaojinensis*. IL represents the one-year post-infected larva of *H*. *xiaojinensis* by *O. sinensis*. ML represents the mummified larva of *H*. *xiaojinensis*.

**Table 1 insects-13-01119-t001:** The list of upregulated genes probably related to *O. sinensis’s* pathogenicity.

Gene ID	log2FC	FDR	Pfam_Annotation	Swiss_Prot_Annotation	Organisms
gene-G6O67_000650	2.55	2.18 × 10^−7^	Cytochrome P450	Cytochrome P450 monooxygenase	*Penicillium roqueforti*
gene-G6O67_004524	2.43	0.000694338	Cytochrome P450	Benzoate 4-monooxygenase	*Aspergillus niger*
gene-G6O67_001204	5.67	2.05 × 10^−6^	Cytochrome P450	Cytochrome P450 52-M1	*Starmerella bombicola*
gene-G6O67_007863	2.50	2.22 × 10^−13^	NADH:flavin oxidoreductase	NADPH dehydrogenase afvA	*Aspergillus flavus*
gene-G6O67_008258	1.73	8.12 × 10^−9^	ESSS subunit of NADH:ubiquinone oxidoreductase (complex I)	--	*Ophiocordyceps sinensis*
gene-G6O67_000430	8.34	1.27 × 10^−14^	Fungalysin metallopeptidase (M36)	Extracellular metalloproteinase	*Aspergillus flavus*
gene-G6O67_002713	1.51	1.89 × 10^−6^	Fungalysin metallopeptidase (M36)	Extracellular metalloproteinase 5	*Arthrodermaotae*
gene-G6O67_007546	1.82	3.44 × 10^−13^	Serine aminopeptidase, S33	Protein bem46	*Schizosaccharomyces pombe*
gene-G6O67_006297	1.61	0.002749971	Serine carboxypeptidase	Carboxypeptidase S1 homolog A	*Trichophyton rubrum*
gene-G6O67_007302	1.82	4.01 × 10^−5^	peptidase_M16	Putative zinc protease mug138	*Schizosaccharomyces pombe*
gene-G6O67_002052	2.53	2.37 × 10^−12^	Redoxin	Peroxiredoxin Asp f3	*Neosartorya fumigata*
gene-G6O67_006384	1.83	8.15 × 10^−7^	CPBP intramembrane metalloprotease	Probable CAAX prenyl protease 2	*Schizosaccharomyces*
gene-G6O67_005828	2.60	7.97 × 10^−6^	Eukaryotic aspartyl protease	Endothiapepsin	*Cryphonectria parasitica*

## Data Availability

The data presented in this study are available on request from the authors.

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
