# Peer review of "Transcriptomic Analysis Insight into the Immune Modulation during the Interaction of Ophiocordyceps sinensis and Hepialus xiaojinensis"

_insects, 2022, doi:10.3390/insects13121119_

Round 1

Reviewer 1 Report (Previous Reviewer 2)

The authors addressed most of my concerns. The below is some more comments for improvements.

In Fig2, FigB Y title overlaps with FigA. You should get more space between two figures.

In Fig2A, move legends "up-regulated", and "down regulated" to the X titles. Move "IL vs. L" somewhere to e.g. upper right as in FigB.

In Fig2B, remove a title "Volcano plot".

In Fig4 No explanation in the figure legend about Diff. Do it for circles and rectangles.

In Fig12 align vertically the larval sample names.

In Fig8 You used too many colors and is too busy. You want to highlight only red and blue triangles, so replace all similar colors used for everything else (e.g. red Fungi, and other blue proteins, blue cell membranes. etc)

In Fig9 Same as Fig2, change X titles. remove "Volcanoe plot"

In Fig11 Same as 4. Explain circles and rectangles in figure legend.

In Fig12 align horizontally and vertically the larval sample names.

Author Response

Reviewer 2 Report (Previous Reviewer 3)

The MS has been revised  as suggested.

Author Response

The MS has been revised  as suggested.

This manuscript is a resubmission of an earlier submission. The following is a list of the peer review reports and author responses from that submission.

Round 1

Reviewer 1 Report

Please re-evaluate the introduction. It doesn't have a cohesive flow and story.

Please go back and review grammar and sentence structures. Some of them don't make sense and need to be edited.

Line 109: How many insects were used for this? This needs to be stated. Were there replicates used in this experiment.

The method sections needs to be revised, there are grammatical errors and typos.

The paper needs to be carefully looked up and the flow of the paper needs to established. 

Author Response

We are very grateful for your precious time in revising our paper and providing valuable comments for the manuscript. According to your advice, we tried our best to amend our article. Thank you very much.

Reviewer 2 Report

Tong et al did the transcriptome analysis of pre-infection, one year post-infection fat bodies, and mummified Hepialus xiaojinensis by Ophiocordyceps sinensis. DEG analysis and the following qPCR found several critical genes that were involved in immune response and fungal pathogenicity. 

I enjoyed reading this manuscript. The experimental design is good, the result has a great novelty, and English is good. However, I have a concern below.

Figure 13, Why RNA-seq bar graph sometimes have error bars but the others don't? Also how many biological replicates do you have for qPCR? Please use scatter plot than the bar graph for visualization. 

Also you should separate qPCR graph from the RNA-seq graph because their "Relative expression levels" units are not identical between graphs assuming normalization methods were different. You can put and align two graphs in the same row so the readers can compare them easily.

Also some comments below.

Figure 1. Can you describe the scale? I assume the left is the centimeters and the right is the inches but please clarify them. Also, which part of the body part in ST was used? I don't think it was described in the main text. Also could you change the abbreviations of CK, IL, and ST to some other terms so readers can easily connect them to pre, post, and mummified larvae unless you have the strong reasons you would like to use these terms? 

Figure 2 and 9. The figure legends (up/down/normal) are too small. Please enlarge them so readers can see the colors.

Figure 5 and 12. "Color Key and Histogram" overlaps with the figure. 

Figure 6. Can you also perform AlphaFold analysis on this protein?

L 163, larger fonts were used. 

L189, the lines got dense.

L386, typo "be play"

Author Response

(The authors gave the same response as above.)

Reviewer 3 Report

The authors take the advantage of the long-term chronic infection of Ophiocordyceps sinensis for studying the immunological interplay between an insect host and a pathogenic fungus. After a comparative transcriptome analysis of CK and IL, as well as  IL and ST, a conclusion was worked out: immune-related genes of the host have no differential expression, and thus the fungal immune tolerance would develop in the late phase of infection in the host,  revealing the mechanism of immune modulation in the long-term chronic infection process between H. xiaojinensis  and O. sinensis. It is well written in English.  How ever, there are still spaces for quality improvement.

1. In general, the authors reveal there experimnetal results mixed with the someone else's, which may result in a confusion to readers.

2. In the notice of fig.3, there is no expaination for "q-value", but in fig.4 it is said that q-value represents adjusted p-value. This is unresonable, the expaination should be put in the first place where it appears, and adjusted p-value is still a p-value.

3. In line 305, ......pathogen recognition __(PRRs), whether one word such as receptors omited?  From line 382 to 394, three Figure 7 should be fig.7A, fig. 7B and fig. 7C, respectively.

4. Thers are many abbreviations, but the abbreviation list is not complete. What is the criterian for selection? 

Author Response

we are very grateful for your precious time in revising our paper and providing valuable comments for the manuscript. According to your advice, we tried our best to amend our article. Thank you very much.

Round 2

Reviewer 1 Report

The authors have corrected everything that I have asked.